# Study of Growth and Properties of Electrodeposited Sodium Iron Hexacyanoferrate Films

**DOI:** 10.3390/ma15217491

**Published:** 2022-10-25

**Authors:** Michael Pohlitz, Christian K. Müller

**Affiliations:** Faculty of Physical Engineering/Computer Sciences, University of Applied Sciences Zwickau, 08056 Zwickau, Germany

**Keywords:** electrodeposition, prussian blue, prussian blue analogs, thin films, resistive switching

## Abstract

Sodium iron hexacyanoferrate (NaFeHCF) films were electrodeposited on Au/Cr/Si for the study of growth behavior and physical properties. The NaFeHCF films were studied by different analytical methods to prove the chemical composition, morphology and crystal structure. The grains of the film grow with a cubic structure with an average lattice parameter of 10.10 Å and the preferential growth direction along the [111] direction of the cubic cell. The films show a repeatable bipolar resistive switching behavior accompanied by high current changes (up to a factor of ~10^5^). The different resistive states in the materials are dominated by ohmic conduction.

## 1. Introduction

Prussian blue analogs (PBA), with the composition A_h_M’_k_[M’’(CN)_6_]_l_·mH_2_O (A corresponds to the alkali metal cation; M’ and M’’ are divalent or trivalent transition-metal cations, respectively; and h, k, l, and m correspond to the stoichiometric coefficients) have been in scope of research for decades [1,2]. These compounds form a face-centered cubic lattice consisting of –NC- M’’-CN-M’-NC units, alkali metal cations as the counterions, and water molecules sitting on the interstitial places. If no or only a fractional number of alkali metal ions exist, water molecules can fill vacant sites and induce distortions of the cubic cell [2].

PBA with M’, M’’ = Fe, named Prussian Blue (PB) has two phases ${\mathrm{Fe}}_{4}^{3+}{\left[{\mathrm{Fe}}^{2+}{\left(\mathrm{CN}\right)}_{6}\right]}_{3}$ (insoluble PB) ${\mathrm{KFe}}^{3+}\left[{\mathrm{Fe}}^{2+}{\left(\mathrm{CN}\right)}_{6}\right]$ (soluble PB) [3]. Due to its outstanding physical properties, PB is in the focus of scientific studies. PBA are good hosts for alkali metal ions [4,5,6] and are of high importance as electrodes in batteries [7,8,9]. For instance, PBA have a high capacity to host sodium ions inside their lattice which can form three-dimensional networks enabling the transport of these ions [10,11,12]. In addition, electrochromism [13,14], thermochromism [15], photochemical magnetism [16], electrocatalysis [17], and sensing [18,19,20] have been shown in PBA. Recently, it was reported when applying a voltage, PB can switch several times between low and high resistive resistance states, known as resistive switching effect [21,22,23]. Thereby, strong evidence was given: that electrical conduction in these materials is dominated by the migration of alkali metal ions. Depending on the stoichiometry of PB, particularly the amount of alkali metal ions, bipolar [21,22] or unipolar resistive switching behavior [23] have been reported.

The fabrication of PBA layers can be performed by electrodeposition [24,25], epitaxial methods [26], spin coating [27], and vapor diffusion [28]. Among them, electrodeposition can be seen as a highly suitable method for the fabrication of PBA because of controlled film parameters (e.g., film thickness, morphology, crystallographic orientation, stoichiometry), high reproducibility, low costs, and fast processing [23]. With careful control of the deposition parameters, electrodeposition can give single-crystalline grains and achieve high crystal quality comparable with the epitaxial method. 

Nevertheless, studies discussing the role of the electrolyte and electrochemical deposition conditions on the PB film characteristics are rare. Additionally, the impact of PBA with altered stoichiometry and different alkali metal ions on the electrical properties, in particular the resistive switching effect, was not sufficiently investigated yet. When changing the type of alkali metal ions and their quantity, the electric transport mechanism could be influenced, resulting in either unipolar or bipolar switching behavior. Moreover, characteristic switching parameters, e.g., switching current, switching voltage, switching speeds, and switching stability, can be influenced to obtain defined device properties. However, up to now such studies with PBA have not been performed. 

Here, we present a work on electrodeposited sodium intercalated PB layers. The impact of sodium ion intercalation in PBA on composition, morphology, structure, and resistive switching behavior is discussed. A bipolar resistive switching behavior with current changes over more than five orders of magnitude was identified. The results are fundamental to understand resistive switching phenomena in Na-rich PBA and to identify their potential for application in electronic devices, e.g., memristors.

## 2. Materials and Methods

### 2.1. Sample Preparation

Potentiostatic fabrication of NaFeHCF layers was realized with a three-electrode setup (Pt counter electrode, saturated calomel electrode (SCE) reference electrode, and 50 nm Au/5 nm Cr on Si(100) as working electrode) as described in ref. [21] at 25 °C. For the electrochemical experiments, a potentiostat (Ivium CompactStat, Eindhoven, Netherlands) was used. All the voltages applied in the electrochemical experiments and cyclic voltammetry refer to the SCE electrode. The distance between the counter and working electrode was set to 20 mm because at this distance the highest current density was observed. The deposition area of the PBA film was ~0.6 cm². As electrolyte for the film preparation a aqueous solution of 0.25 mM K_3_Fe(CN)_6_ (ACS, >99%, Sigma Aldrich, Darmstadt, Germany), 0.25 mM FeCl_3_ (ACS, 98–102%, Sigma Aldrich, Darmstadt, Germany), 1.0 M NaCl (ACS, 99–100.5%, Sigma Aldrich, Darmstadt, Germany) and 5.0 mM HCl (ACS, 37%, Sigma Aldrich, Darmstadt, Germany) at pH 2 was used [25]. The deposition potentials were determined from cyclic voltammetric measurements (Figure 1). The oxidation peak related to the formation of NaFeHCF was observed at ~0.29 V. This voltage was used to grow the NaFeHCF films. The deposition process was finished when achieving a deposited charge of 30 mC.

### 2.2. Sample Analysis

The morphology and elemental composition of the samples were studied by field emission scanning electron microscopy (FEG-SEM, TESCAN CLARA, Brünn, Czech Republic) equipped with an energy-dispersive X-ray (EDX) detector (Ultim Max 65 SDD, Oxford Instruments, Wiesbaden, Germany) at 10 kV. In addition, the chemical composition of the samples was analyzed by Raman spectroscopy (Witec RISE, Ulm, Germany) performed inside the electron microscope. The Raman measurements were performed with a 532 nm laser at 1 mW. 

The surface elemental composition and oxidation state of the elements in our samples were determined by X-ray photoelectron spectroscopy (XPS, SPECS, Berlin, Germany). Al Kα radiation was used as the excitation source. The size of the investigated area of interest from the surface was ~1 mm. The spectral analysis was performed with Casa (CasaXPS, vs. 2.2.24, USA).

The crystallographic structure of the samples was studied with a Siemens X-ray diffractometer (D5000, Siemens, München, Germany) using Bragg–Brentano geometry, a Cu Kα X-ray source (λ = 1.5418 Å), and a scintillator detector. A slit size of 1 mm was used for the incident and the diffracted X-ray beam to limit the measured area of the sample to the PBA film. 

Electrical measurements at room temperature were performed with a Keithley 2401 source meter (Keithley, Cleveland, OH, USA). For the electrical two-point measurements, the Au layer was used as the bottom contact and conductive silver paste on top of the PBA film was used as the top contact.

## 3. Results and Discussion

Figure 2 summarizes the results from the SEM surface analysis of NaFeHCF. The surface images (Figure 2a,b) show complete and homogeneous coverage of the substrate with grains. The surface morphology of NaFeHCF shows triangular pyramidal grains with a typical size (edge length) ranging from 100 to 500 nm. Most of the grains are orientated with the pyramids’ corners perpendicular to the substrate surface. However, some grains show slight deviations from this preferential direction. A similar orientation was found for PBA deposited at lower voltages (0.1 V) [24]. 

Figure 2c shows a cross-sectional analysis of the NaFeHCF layer. The mean film thickness amount to 800 nm. A columnar microstructure is visible along the whole layer, whereas the columnar grains have grown from the Au/PB interface up to the PB surface (see Figure 2b). It seems that the grains near the substrate are smaller than those close to the top, indicating a Volmer–Weber growth process [21]. In addition, in the case of NaFeHCF, the appearance of cracks was observed during electron microscopy imaging (see Figure 2a). This crack formation can be an indication for significant amount of coordination water molecules within the cubic structure. Irradiation with the electron beam can induce excitation of these water molecules and consequently result in a strain gradient allowing breakage of the layer. Moreover, thermal expansion of differently oriented grains can induce additional strain in the film. 

When investigating the crystalline structure with X-ray diffraction, a predominant growth direction of the crystallites parallel to the [111] orientation (out-of-plane direction) is observed (Figure 3). However, deviations from this preferential crystallite orientation can also be found because of the presence of (200) and (220) reflexes. Assuming a cubic structure, based on the (111) and the (222) reflexes, appearing, respectively, at 2θ = 15.25° and 2θ = 30.53°, a lattice constant of a = 10.10 Å can be calculated. This value is between the cubic lattice parameters reported for PW (high amount of alkali metal ions, a = 10.05 Å) and PB (without alkali metal ions, a = 10.20–10.25 Å [24], for comparison PDF Card No. 01-074-9174). Whereas, the authors of [28] reports a value of a = 10.16 Å for PB, in good agreement with our value. The NaCl peak comes from small amounts of the electrolyte on the sample surface. Typically, most of the NaCl is located at the border of the NaFeHCF film.

EDX-analysis was applied to analyze the layer composition (see Figure 4). A typical EDX spectrum taken from the sample surface is shown in Figure 4a. The elements show a homogenous distribution over the surface (Figure 4b). Line-scan analysis along the cross-section of the NaFeHCF film (Figure 4c,d) gave 42 at% C, 34.1 at% N, 14 at% O, 8.5 at% Fe, and 1.4 at% Na. From this composition, a mixture of 33% NaFe^3+^[Fe^2+^(CN)_6_]∙mH_2_O and 67% Fe^3+^_4_[Fe^2+^(CN)_6_]_3_∙mH_2_O can be proposed due to nearly stoichiometric C/N ratio (~1.23) and high Fe/Na ratio (~6). The amount of C is slightly higher than N because of carbon surface contaminations. It is important to note that also a high amount of O was measured, strongly indicating the presence of coordination water, and identified as a source of strain under electron beam heating. The average number of coordination water molecules can be estimated to be m = 3 when taking into account the measured oxygen content and the proposed stoichiometry.

In addition, XPS analysis was used to further investigate the surface composition in our samples. As indicated in Figure 5a, the samples contain Fe, C, N, Na, and O, confirming the presence of NaFeHCF. The C1s peak (appearing at 284.6 eV) was the reference [29]. A small amount of Na can be observed, proving the formation of NaFe[Fe(CN)_6_] beside Fe_4_[Fe(CN)_6_]_3_ in the surface region of the layer. Figure 5b presents a detailed view from the Fe2p core level region. The XPS peak of the $\mathrm{Fe}2p_{3/2}$ and $\mathrm{Fe}2p_{1/2}$ states with their corresponding curve fits prove the presence of ${\mathrm{Fe}}^{2+}$(fit maxima at 708.5 eV and 721.0 eV) and ${\mathrm{Fe}}^{3+}$ (fit maxima at 710.5 eV and 723.0 eV) in our samples. In addition, two satellite peaks can be identified (maxima located at 714.5 eV and 727.0 eV). These findings are in agreement with the reduction of Fe^3+^ to Fe^2+^ ions that occurs during the electrochemical fabrication of NaFeHCF. Figure 5c shows the O1s core level region. The peak deconvolution gives two peaks located at 535.6 eV and 531.9 eV, respectively. The first peak could be a fingerprint for water and supports the assumption from EDX analysis. The second peak probably originates from metal oxides. However, to understand these results, a deeper analysis needs to be performed. 

Raman spectroscopy was used to extract further details about the chemical composition of the samples. Figure 6 shows a typical Raman spectrum obtained on the NaFeHCF film. The Raman spectrum clearly indicates the formation of NaFeHCF. The CN groups bond to ${\mathrm{Fe}}^{2+}$ and ${\mathrm{Fe}}^{3+}$ in case of NaFeHCF show a stretching vibration in the range between 2200 and 2050 cm^−1^. When analyzing the peak fine structure a strong peak at 2155 cm^−1^ for CN groups attached to Fe(III) can be seen. This peak is assigned to the 1A_g_ ν(CN) stretching vibration [30]. The small shoulder at 2130 cm^−1^ is a fingerprint for CN^−^ ions [31]. A latter signal was also observed in soluble forms of PB with a significant amount of Na ions [32]. Another peak appearing at 2095 cm^−1^ refers to the CN groups attached to Fe(II). This peak is in literature assigned to the E_g_ ν(CN) stretching vibration [33]. Other peaks at wavenumbers < 650 cm^−1^ of the spectrum refer to the stretching vibration of Fe–C and the deformation vibration of Fe–CN–Fe [34]. Important to note is that the Raman peaks have an FWHM of ≥10 cm^−1^, which probably originates from strain [35], supporting our results from electron microscopy. Additional peaks with low intensity can be seen at wavenumbers >2400 cm^−1^ and in the range from 600 to 1100 cm^−1^. The origin of these signals is not clearly understood and probably could be a sign for carbon sources or oxides in our samples. Important to note is that NaCl found in XRD was not present in the investigated Raman region (area in the center of the sample).

Figure 7a shows a typical electrical current-voltage plot in a semi-log scale of our samples using an Au/NaFeHCF/Ag layer setup. The measured I–V behavior with a butterfly-shaped characteristic is typical for bipolar resistive switching devices [36,37]. When applying a potential cycle with the sequence 0 V→+2.0 V→0 V→−2.0 V→0 V, the “set” and “reset” current changes can be observed at positive and negative bias voltages. Initially without an applied voltage the layer is in a high resistance state (HRS). The set process accompanied by the switching into the low resistance state (LRS) occurs at 0.85–0.95 V (see Figure 7b). When returning the bias voltage to zero and moving to negative voltages, the layer switches back to the HRS at −0.55 to −0.75 V. The measured current range spans values from 2 × 10^−4^ mA to 10^−11^ A. Current changes higher than five orders of magnitude can be observed at the switching voltages and demonstrates the high potential as non-volatile random access memory device. In this work, an enhanced change of switching current was observed, higher than in our earlier work [22], indicating the important role of alkali metal ions in resistive switching devices. The sodium ions have lower mobility and a lower conductivity compared with potassium ions. Therefore, in the case of sodium ions, the resistance change, measured before and after the formation of ionic conductive filaments and probably responsible for resistive switching, can be higher. In addition, because of the higher layer thickness of the NaFeHCF films, the resistance in HRS can achieve higher values explaining the greater current drop between LRS and HRS.

Figure 7b presents cyclic measurements of the Au/NaFeHCF/Ag device. Generally, similar switching behavior and curve shape were observed during the investigated number of cycles. Therefore, we can state that the resistive switching is reversible in this material. However, the switching voltages change slightly from cycle to cycle for both the set and reset processes. We believe that changes in the conductive filament formation can be the reason for the different switching voltages. The used silver paste on top of the film covers a large area (typically 50 × 50 µm²) of the film. When applying a voltage, there are different possibilities in cross-section to form conductive filaments involving grains and grain boundaries. Consequently, the set and reset values can vary.

In Figure 8, the conduction behavior of the NaFeHCF film is analyzed for several cycles in the positive bias voltage range. For LRS and HRS, a slope near 1 presents an ohmic behavior. The latter was found to be characteristic for conduction through filaments [38,39]. Most of the calculated values are in the range from 0.63 to 1.27 associated with ohmic conduction. Nonlinear behavior with a higher slope (>2), as observable for cycle 5 at >0.3 V, could be attributed to other processes of conduction e.g., space charge limited current (SCLC) mechanism [40]. 

## 4. Conclusions

Homogeneous layers of NaFeHCF with a thickness of ~800 nm were obtained by potentiostatic electrodeposition. An electric potential of 0.29 V was found to be useful to obtain high-quality films with defined crystal structure, crystallographic orientation, and composition as confirmed by electron microscopy, X-ray methods, and optical spectroscopy. The Au/NaFeHCF/Ag structure showed a repeatable bipolar switching behavior with high current changes (up to 10^5^). The measured electrical currents are typical for the ohmic conduction model. Further investigations will be performed to obtain a deeper understanding of the influence of fabrication parameters on electrical properties. In particular, the study of the underlying conduction mechanism and the long-time electrical behavior in this material will be the aim of future work.

## Figures and Tables

**Figure 1 materials-15-07491-f001:**
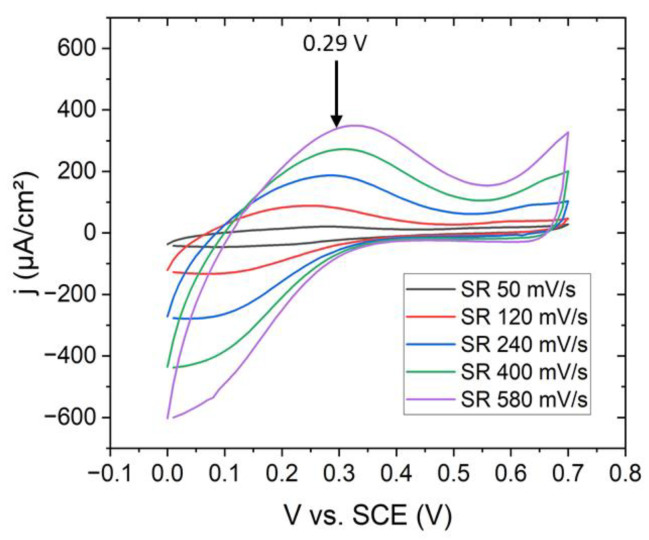
Cyclovoltammetric curves at different scan rates on Au/Cr/Si(100) substrates for the deposition of NaFeHCF.

**Figure 2 materials-15-07491-f002:**
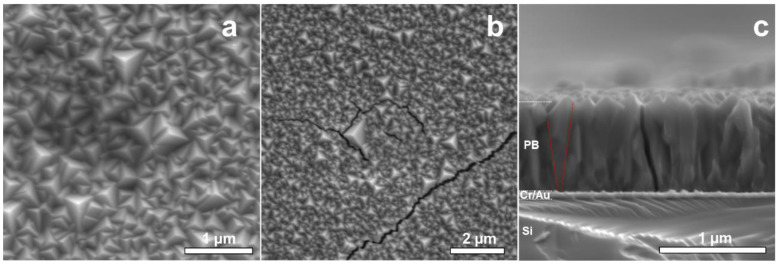
SEM images of the NaFeHCF layer electrodeposited at 0.29 V: (**a**) Surface view of NaFeHCF, (**b**) Crack formation in NaFeHCF layers by electron beam irradiation, (**c**) cross-section view. The red lines in (**c**) mark the change in column size from the top of the film to the bottom.

**Figure 3 materials-15-07491-f003:**
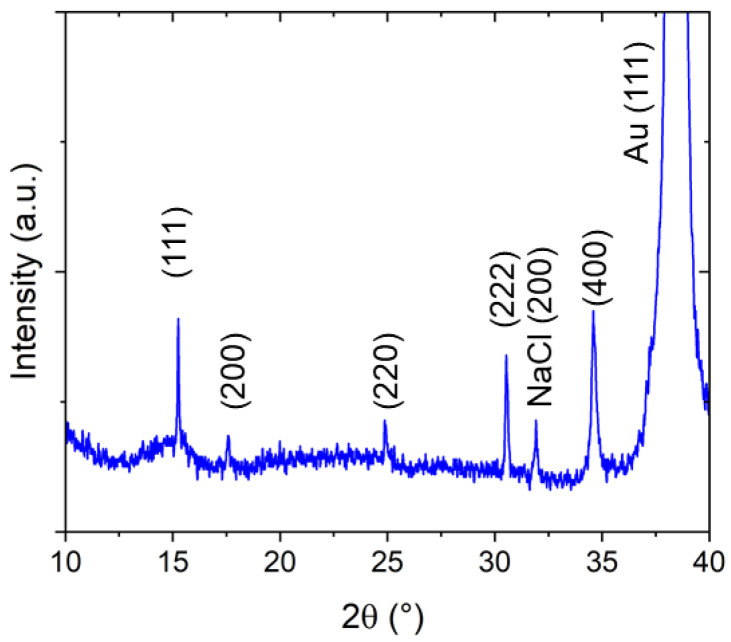
X-ray diffraction pattern of NaFeHCF film grown at ~0.3 V measured with Cu Kα radiation.

**Figure 4 materials-15-07491-f004:**
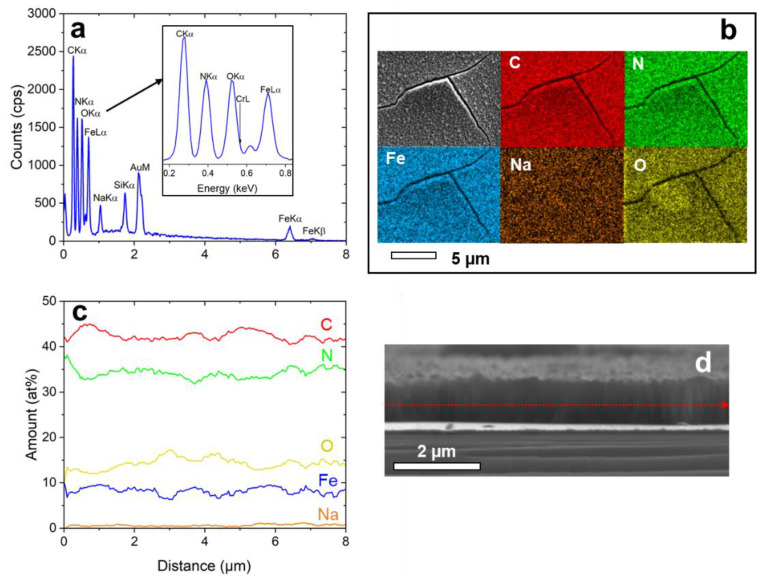
EDX-analysis of NaFeHCF measured at 10 keV: (**a**) EDX spectrum taken from the area in (**b**), (**b**) elemental maps of the homogeneously distributed elements, (**c**) Elemental composition obtained by line-scan analysis along the marked line in (**d**), (**d**) SEM cross-section image of the investigated sample region used for line-scan analysis.

**Figure 5 materials-15-07491-f005:**
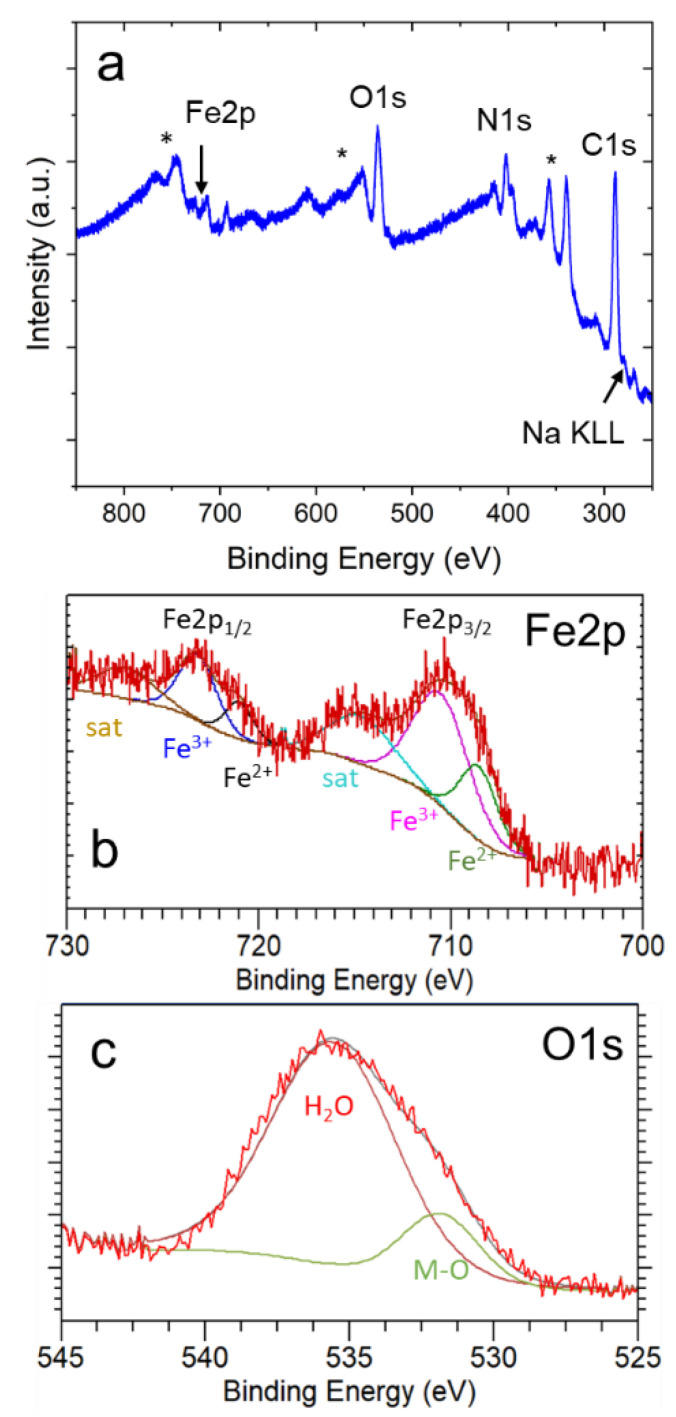
XPS results obtained from the surface of electrodeposited NaFeHCF layers: (**a**) overview spectrum, (**b**) detailed view of the Fe2p core level region, (**c**) detailed view of the O1s core level region. The stars in (**a**) mark the signals from the Au/Cr/Si substrate.

**Figure 6 materials-15-07491-f006:**
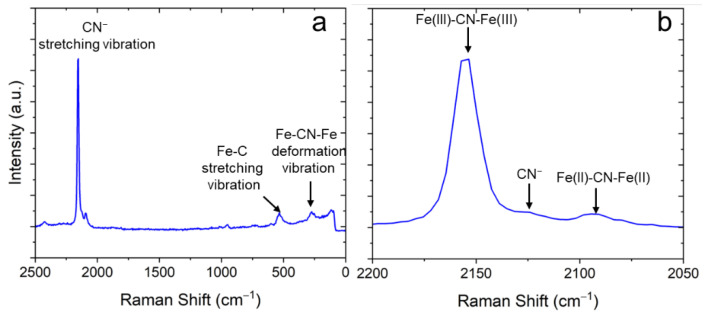
Typical Raman spectrum of electrodeposited NaFeHCF layers measured with 532 nm laser at 1 mW: (**a**) overview, (**b**) detailed view of the spectral region around the CN-stretching vibration.

**Figure 7 materials-15-07491-f007:**
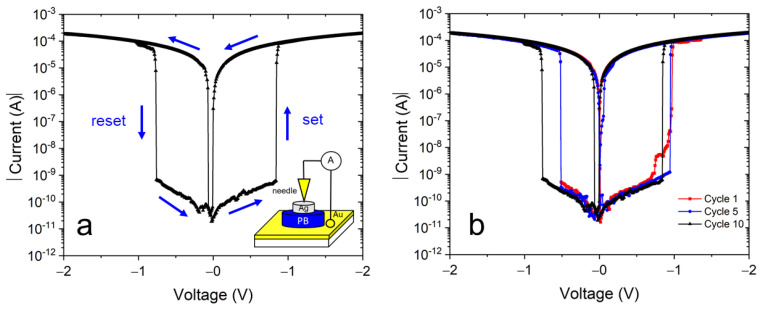
(**a**) Typical I–V curve of the Au/NaFeHCF/Ag layer sequence. The inset shows the used measurement geometry. (**b**) I–V curves of Au/NaFeHCF/Ag layers at different cycle numbers. The bias voltage range was +2.0 V to −2.0 V.

**Figure 8 materials-15-07491-f008:**
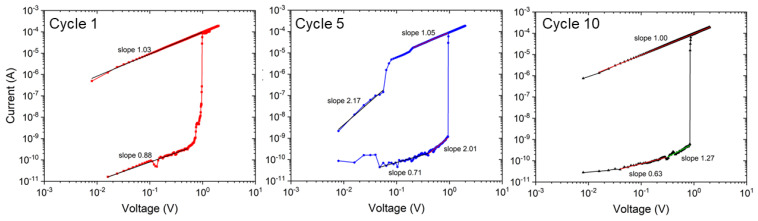
Logarithmic current-voltage plots in the positive bias voltage range of the electrical measurements for cycle number 1, 5, and 10.

## Data Availability

Not applicable.

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
