# Peer review of "Study of Growth and Properties of Electrodeposited Sodium Iron Hexacyanoferrate Films"

_materials, 2022, doi:10.3390/ma15217491_

Round 1
Reviewer 1 Report
Congratulation on your work, focused on electrodeposited NaFeHCF films. My opinion is that, in its current form, the paper needs major revisions before publication. Please find below my suggestions that, I hope, can improve the overall quality of the paper.
Introduction.
Please avoid block citation (e.g., [4-9]), as citations should be described in details. Please use maximum 3 citations together.
The motivation of the work is not clearly stated, only briefly. The authors should discuss the gap that this study should fill. What is the motivation and novelty of their work? Which materials properties are the authors aiming to improve as a result of their study?
Sample Analysis
Please introduce the abbreviations before their first use (e.g., EDX).
XRD: please give more information on the optics used (open or closed detector configuration, slits etc.). Slits dimension determines the signal intensity and the width of the diffraction peaks, but also, in case of a close detector configuration (narrow slits) the probability of small intensity peaks to be masked, giving the possibility of wrong „preferential” orientation conclusion.
Results and Discussion
The interpretation of the films’ morphology from SEM data, as discussed within the text, does not fit with the presented images. Figure 2 shows star-like or triangle like grains, not cubic. Please rewrite the entire first paragraph from Results and Discussion section.
Cross-section discussion (Figure 2c). The same observation: “The morphology with cube grains is visible …”. In cross- section the microstructure can be seen, not morphology. Also, the film shows a columnar growth; in the presented images no “cube grains” can be seen. For clarity, please mark in Figure 2c the composition of each layer: the Si substrate, the Cr and Au buffer layers, and the deposited film.
The authors conclusion regarding the grains size at the interface, compared with the one on top of the film cannot be drawn from the presented SEM image. In order to avoid cracks formation during SEM imaging, did the authors have the option to use low vacuum mode (or a Peltier stage), for analysis, in order to reduce the probability of crack formation?
3rd paragraph. The XRD can give info on microstructure and phase composition, but not on morphology. Please correct.
Please give info on the PDF card no. (data base) used for cell parameters, as reference. The peaks marked as “substrate” are coming from buffer layers (Au and or Cr) or from Si substrate? Please clarify on the Figure. Please mark the orientation (hkl) of the NaCl impurity. Was XRD data collected at grazing incidence (as suggested by the used 2 theta axis)? If so, please specify the omega angle value.
EDS data. My suggestion to the authors is to mark the elements where they mention the substrate, e.g., Kalpha line for Si, at 1.739 keV, and M line for Au, at 2.1 keV. Surprisingly, there are no Cr Lalpha and Kalpha lines seen in the EDS spectra; it should be detectable, even though the Cr layer is only 5 nm thick. Can the authors introduce a zoom area for low energy range, for ex. 0-2 keV, to better visualize this energy region?
Regarding EDS. As the probed volume by the e-beam is depth and e-beam energy dependent, a better representation of the films elemental composition would have been given by scan-lines on cross-sections. In this way, a thickness distribution of the elemental composition could be obtained, giving a better image of the films’ composition homogeneity.
From EDS data, the authors determined, indirectly, the presence of water molecules. As EDS and XPS cannot directly detect hydrogen, the authors should have used XPS to determine if the detected oxygen is coming from an impurity phase or from H2O and, therefore, used XPS as a direct proof of water formation.
Raman results. The XRD data indicated the presence of NaCl, as impurity, while the authors concluded, from Raman spectra, that there are no impurities. The authors should clarify this contradiction in experimental data. How relevant is the presence of NaCl to the observed crack formation within the films?
In Figure 6a there are regions (between 2500-2250 cm-1 and ~1200-600 cm-1) with peaks which are not attributed to any composition. Are these peaks coming from impurity phases?
Figure 8. Can the authors explain the mechanism given the difference in IV curves between cycle 5 and cycles 1 and 10? Could you please explain the mechanism for reverting, at cycle 10, at a similar behavior as for cycle 1? Please explain, in text, the used SCLC acronym.
References.
Please carefully check the reference list and their correspondence within the manuscript text. There are some mixed up lines within the references (e.g., Ref. 5, 30, 32, 37).
Good luck!
Kind regards.
Reviewer 2 Report
The paper studied the growth and properties of electrodeposited sodium ion hexacyanoferrate films. The as-prepared NaFeHCF films were characterized by SEM, XRD, EDS, XPS, and Raman spectrum. Interestingly, the film showed a repeatable bipolar resistive switching behavior accompanied by a high current change. The paper provides some interesting finds but needs major revision before being published in the journal.
1, “The deposition area of the PBA film was ~0.6” in the sample preparation section. What is the unit?
2, electron microscopy should be written as scanning electron microscopy for clarity.
3, In page 6, “higher than in our earlier work [21]”, ref. [21] should be 22.
4, The full name of SCLC should be given.
5, Do the thickness of the films affect the switching behavior and the current ratio? Why?
6, What are the reason for the shifts of the set/reset voltages? Is it possible to have stable set/reset voltages?
7, How about the current ratio changes under a high number of cycles, such as 100 cycles?
Reviewer 3 Report
The authors have presented the paper entitled "Study of growth and properties of deposited sodium iron hexacyanoferrate films". The general structure and composition of the paper is good and the topic is of interest for the audience. However some issues have to be addressed before publishing the article.
Page 2 line 65 deposition area about 0.6 (no units are given)
Page 3 lines 99 – 101 “Most of the cubes are orientated with the crys- 99 tallographic [111] direction perpendicular to the substrate …” How can the authors be sure of the orientation just based on the micrograph? Does any EBSD was performed?
If any other orientations are present , those orientations could cause stress and/or texture in the film?
Page 3 lines 112 & 113; I believe the authors refer to Figure 2b instead of 2a, as stated in the text. Moreover, cannot this crack be due to stress in the sample caused by the different grain orientations? Especially, considering the 220 and 200 orientations, together with the presence of the NaCl crystallites.
Page 4 Figure 4a. I strongly recommend that in the EDX spectra peaks must be labelled with the corresponding element and perhaps sub labelled with another name as for “substrate”. Because, it is not correct to label as “substrate” some peaks without the corresponding element.
Page 5 Figure 5b. The XPS peaks corresponding to Fe2p3/2 and Fe2p1/2 seems a bit shifted for the +2 and +3 corresponding to metallic Fe. Moreover, due to the presence of Oxygen, Have you check the O1s peak and try to solve if there is any iron oxides? The satellite also seems to big, perhaps another oxidation state?
Can the authors show the other deconvolutions and try to correlate?
I hope the authors can provide proper comments and response to these questions
Round 2
Reviewer 1 Report
The authors made the suggested corrections to the manuscript. I believe that in its current form the manuscript can be published in the Materials journal.
Reviewer 2 Report
After revision, the manuscript has been improved a lot. I recommend publishing it.